# Establishment of an Efficient Sugarcane Transformation System via Herbicide-Resistant CP4-EPSPS Gene Selection

**DOI:** 10.3390/plants13060852

**Published:** 2024-03-15

**Authors:** Wenzhi Wang, Talha Javed, Linbo Shen, Tingting Sun, Benpeng Yang, Shuzhen Zhang

**Affiliations:** 1National Key Laboratory for Tropical Crop Breeding, Institute of Tropical Bioscience and Biotechnology, Chinese Academy of Tropical Agricultural Sciences, Haikou 571101, China; wangwenzhi@itbb.org.cn (W.W.); talhajaved@itbb.org.cn (T.J.); shenlinbo@itbb.org.cn (L.S.); suntingting@itbb.org.cn (T.S.); yangbenpeng@itbb.org.cn (B.Y.); 2Sanya Research Institute, Chinese Academy of Tropical Agricultural Sciences, Sanya 571763, China; 3Crop Genomics and Bioinformatics Center and National Key Laboratory of Crop Genetics and Germplasm Enhancement, Nanjing Agricultural University, Nanjing 210095, China; 4Hainan Yazhou Bay Seed Laboratory, Sanya 571763, China

**Keywords:** sugarcane, *Agrobacterium*-mediated transformation, transgenic technology, herbicide resistance

## Abstract

Sugarcane (*Saccharum* spp.), a major cash crop that is an important source of sugar and bioethanol, is strongly influenced by the impacts of biotic and abiotic stresses. The intricate polyploid and aneuploid genome of sugarcane has shown various limits for conventional breeding strategies. Nonetheless, biotechnological engineering currently offers the best chance of introducing commercially significant agronomic features. In this study, an efficient *Agrobacterium*-mediated transformation system that uses the herbicide-resistant CP4-EPSPS gene as a selection marker was developed. Notably, all of the plants that were identified by PCR as transformants showed significant herbicide resistance. Additionally, this transformation protocol also highlighted: (i) the high yield of transgenic lines from calli (each gram of calli generated six transgenic lines); (ii) improved selection; and (iii) a higher transformation efficiency. This protocol provides a reliable tool for a routine procedure for the generation of resilient sugarcane plants.

## 1. Introduction

Sugarcane (*Saccharum* spp.) is one of the most important industrial crops, contributing significantly to economics and food security [1]. Sugarcane is primarily grown as a source of sugar and bioethanol, meeting around 80% and 40% of the global sugar and bioethanol demands, respectively [2]. Climate change adversely affects both the quantity and quality of many crops worldwide. Higher temperatures will exacerbate drought stress and accelerate crop development, as well as increase crop yield variability and the likelihood of pathogen-related yield failures [3]. Additionally, sugarcane productivity is vulnerable to climate change through the direct and indirect effects arising from herbicides [4]. Herbicides may have an impact on the prevalence or severity of infections as well as plant pathogens. Indirect herbicide effects on soil microorganisms that are often hostile to root infections may exacerbate plant diseases. Certain herbicides have the potential to directly harm crop plants, impairing their ability to fend off disease and increasing their susceptibility to infections. When plants sustain sublethal injuries, their root tissues’ membrane permeability may change, allowing the cell contents to seep into the root–soil interface [5,6]. A number of the effects of herbicides, including the effects of glyphosate-based herbicides, and herbicide-resistant plants are reviewed by Schütte et al. [6]. Therefore, in the current period of climate change, the creation of sugarcane cultivars that are multi-stress-resistant has become a research hotspot [7,8,9].

Advances in plant-breeding-based technologies have contributed significantly to sugarcane improvement in recent decades [10,11,12]. For instance, sugarcane breeders interested in increasing herbicide resistance crossed *Saccharum officinarum* (2n = 8x = 80; x = 10) with a wild and vigorous relative, *S. spontaneum* (2n = 5 − 16x = 40 − 128; x = 8), and then backcrossed the hybrids with *S. officinarum*. The hybridization of both genera resulted in modern cultivars with chromosome numbers ranging from 110 to 130, of which 80% originated from *S. officinarum* and 10–15% from *S. spontaneum*, with about 5–10% being recombinant chromosomes [1]. The complex genome and poly/aneuploidy nature of sugarcane as well as the current climate change scenarios complicate agronomic trait improvement through traditional breeding strategies [1]. Notably, transformation and transgenic techniques are now being applied routinely to diverse plant species, especially sugarcane [13,14,15,16,17]. Furthermore, the application of transgenic technology depends on the creation of an effective embryogenic system. Biotechnologies like cell and tissue culture and genetic transformation techniques play a major role in boosting the yield of sugarcane cultivars. Numerous techniques for plant regeneration in sugarcane have been explored to date, including direct shoot organogenesis [12,13,14]. Specifically, the *Agrobacterium-tumefaciens*-mediated transformation of commercial sugarcane cultivars is a more desirable approach due to the ease of application and low cost as well as its relatively higher efficiency compared to other techniques [14].

The efficiency of sugarcane transgenesis can be improved by adopting appropriate transgenic selection systems, such as PMI/Mannose, CP4-EPSPS/glyphosate, and bar/Basta selection [13]. For instance, Wang and colleagues reported on the *bar*/Basta sugarcane transformation system. In this system, selection efficiency (SE = PCR-positive shoots/resistant shoots) was close to 100% [13]. Similarly, Baso and colleagues established a *Bar* gene transformation system against a glufosinate-ammonium herbicide. The SP80-3280 sugarcane cultivar was effectively transformed using this technique, which included reduced oxidation and recalcitrance, a high production of embryogenic calli, enhanced selection and shoot regeneration, and the rooting of the altered plants. These enhancements were combined to produce a transformation efficiency of 2.2%. Another study also highlighted that the transgenic sugarcane resistance to the Basta herbicide obtained by *Bar* gene screening had a lower application prospect [14,15,16]. Conversely, several reports suggested the use of the CP4-EPSPS system for a higher resistance to the Roundup herbicide and vast economic value [14,15,16]. However, little is known about the CP4-EPSPS transgenic sugarcane transformation system. Therefore, an *Agrobacterium*-mediated transformation system that uses the CP4-EPSPS gene against the Roundup herbicide as a selection marker for sugarcane is developed in this study.

## 2. Results

### 2.1. Establishment of Embryogenic Callus

From the immature stalks of sugarcane, explant of about 5 cm in length containing the immature leaf whorl was cut (Figure 1a). After disinfection, the immature top stalk was cut into 1 mm thick transverse sections and transferred to the callus induction medium (Figure 1b). After 25–30 days of culture, mucilaginous calli formed on the transverse sections (Figure 1c). At 40–45 days prior to the herbicide assay and transformation, healthy embryonic calli (light yellow) were selected and cleaned, and the excess culture medium was washed away (Figure 1d).

### 2.2. Determination of Minimum Inhibitory Concentration of Roundup

Compact calli were transferred to Petri dishes containing different concentrations (0 mg/L, 10 mg/L, 20 mg/L, 30 mg/L, 40 mg/L, 50 mg/L) of glyphosate for two weeks followed by transfer to regeneration medium containing the abovementioned same concentrations of glyphosate for an additional two weeks (Figure 2). Notably, the regeneration of putative plants decreased with the increase in glyphosate concentration. Plant regeneration in Petri dishes with 0 mg/L and 10 mg/L of glyphosate resulted in a remarkably high rate. The regeneration of the callus started to be reduced significantly in Petri dishes containing 20 mg/L glyphosate. Notably, only a few calli regenerated with fine buds in Petri dishes with 30 mg/L glyphosate. Additionally, no plant regeneration occurred in Petri dishes containing 40 mg/L and 50 mg/L of glyphosate (Figure 2). Overall, the lethal dose of glyphosate for sugarcane callus was 40 mg/L.

### 2.3. Selection of Resistant Shoots

A total of three Petri dishes, each containing 3.5 g of embryogenic calli (air dried), were prepared prior to transformation (Figure 3a). After *Agrobacterium* induction, co-cultivation, and recovery cultivation, the infected calli were transferred to the selection mediums containing 20 mg/L, 30 mg/L, and 40 mg/L of glyphosate, respectively. After 35 days of cultivation in the dark, most of the transformed calli died, and a few calli sprouted which showed significant resistance to glyphosate (Figure 3b). Resistant calli were transferred to a regeneration medium containing the same glyphosate concentrations and cultured for 20 days (Figure 3c). All of the regenerated shoots were transferred to elongation medium and cultured for 20 days followed by transfer to rooting medium (Figure 3d).

### 2.4. Green Fluorescent Protein Expression

The infected calli after three days of co-cultivation, resistant calli after 30 days of selection cultivation, and resistant shoots after rooting cultivation were evaluated for GFP expression under a microscope. Bright green fluorescence was observed on the infected calli, indicating a high level of *Agrobacterium* infection. Resistant calli showed bright green fluorescence in comparison with non-resistant calli. Bright green fluorescence on resistant shoots, particularly leaves and roots, indicated an absence of chimera transgenic shoot (Figure 4).

### 2.5. PCR Assay of Potential Transformed Shoots

After rooting cultivation, all the resistant shoots were evaluated for *CP4-EPSPS* and *GFP* gene integration by PCR. Notably, *CP4-EPSPS* and *GFP* genes were PCR negative for wild-type sugarcane genomic DNA. All the 21 resistant shoots from 40 mg/L glyphosate selection were PCR positive for the *CP4-EPSPS* gene, while 15 shoots among them were PCR positive for the *GFP* gene. *Agrobacterium* absence in the resistant plants was assessed by *VirG* gene PCR detection with primers specific to the *VirG* gene. *Agrobacterium* was not detected in the resistant sugarcane plants from the rooting medium, as evidenced by the lack of amplification in those plants. It is noteworthy that only PCR-positive, resistant plants were transplanted to the greenhouse (Figure 5).

### 2.6. Transformation Efficiency

The number of resistant shoots from 3.5 g of calli under 20 mg/L, 30 mg/L, and 40 mg/L glyphosate application was 31, 19, and 21, respectively. The highest number of resistant shoots was obtained under a low concentration of glyphosate, 20 mg/L (31 shoots). Notably, 29 shoots were positive for *CP4-EPSPS* with 93.4% selection efficiency (SE). Additionally, SE under 30 mg/L and 40 mg/L glyphosate was 100%. In terms of the transformation efficiency (TE), the number of shoots/weight (gram) of the calli that were PCR positive for the *CP4-EPSPS* gene at the 20 mg/L concentration was 7.4 lines/gram, followed by 4.9 lines/gram at 30 mg/L and 4.2 lines/gram at 40 mg/L. Overall, a low concentration of glyphosate showed lower selection efficiency but higher transformation efficiency (Table 1).

### 2.7. Detection of CP4-EPSPS Gene Expression by Quick Stix Strips

*CP4-EPSPS* gene expression of 21 transgenic shoots from the 40 mg/L glyphosate selection was tested by Quick Stix Strips (Genesino (Dalian) Biological S&T Development Co. Ltd. Dalian, China). Testing lines appeared on each Stix strip for each of the 21 transgenic shoots. The results were in line with the PCR and showed 100% positive testing of *CP4-EPSPS* gene expression in the 21 transgenic shoots from the 40 mg/L glyphosate selection (Figure 6).

### 2.8. Herbicide Sensitivity

The herbicide sensitivity of the transgenic shoots was tested by spraying different concentrations of glyphosate (0%, 0.2%, 0.4%, 0.6%, and 0.8%). After 10 days of spraying different concentrations of glyphosate, all wild-type sugarcane tillers died, while transgenic sugarcane seedlings grew healthily even when treated with up to 0.6% glyphosate. Growth of the transgenic sugarcane seedlings was apparently affected when treated with 0.8% glyphosate (Figure 7).

## 3. Discussion

*Agrobacterium*-mediated transformation is the most widely used and genetically stable technique for transferring T-DNA expression constructs into plant genomes. For low-copy expression, transgenic plants that undergo *Agrobacterium*-mediated transformation are mostly genetically stable [17,18,19,20,21,22,23,24,25]. In the previous research of our lab, two sugarcane transformation selection systems, PMI/Mannose [26] and the *Bar*/Basta selection system [13], were established. Both of these transformation systems had high efficiency due to high-quality sugarcane embryogenic callus induction and a suitable *Agrobacterium*-mediated transformation protocol. High-quality embryogenic callus induction is the key factor for the success of sugarcane genetic transformation. By modifying the 2,4-D concentration in different stages of callus induction, a high quality of sugarcane embryogenic calli was obtained and used for glyphosate killing curve testing and for genetic transformation. *CP4-EPSPS*/glyphosate transformation selection systems have been used in the genetic modification of other crops, such as wheat, cotton, soybean, and corn [27,28,29,30]. In this research, by referring to the selection concentration of glyphosate used in the genetic transformation process of maize, the killing curve for the testing of callus cultivation and regeneration cultivation was used, and a candidate concentration of glyphosate for sugarcane genetic transformation was selected. The SE of 20 mg/L, 30 mg/L, and 40 mg/L glyphosate was 93%, 100%, and 100%, which indicates that a higher concentration of glyphosate obtains higher selection efficiency. However, the TE was 7.4 lines/gg, 4.9 lines/g, and 4.2 lines/g, respectively, with transformation with 20 mg/L glyphosate selection obtaining the highest TE. So, 20 mg/L of glyphosate modified in the selection medium was the best concentration for sugarcane transformation using the *CP4-EPSPS* gene as the selection agent. Our research results are consistent with those of previous reports which also obtained a similar trend for TE [13,19].

Using a fluorescence reporter gene to assist in the establishment of a new genetic transformation selection system is helpful. In this research, we observed the fluorescence of GFP reporter genes throughout the transformation process to assay the quality of the embryogenic calli and efficiency of *Agrobacterium* infection protocol. Through the fluorescence from a single site, the formation process of the resistant calli can be observed. A total of 19 and 21 transgenic shoots from 30 mg/L and 40 mg/L glyphosate selection had positive PCR and Quick Stix Strip assay results, indicating that, under higher pressure of glyphosate selection, only the shoots with *CP4-EPSPS* gene integration and expression can survive throughout the selection process. The number of PCR lines positive for *GFP* gene was less than that of the selectable marker genes, mostly because of the fracture of and missing target gene during the process of integration. The fracture and lack of the target gene during transformation also happened in our previous research [13]. In this study, 10–30% of the selected lines lacked the transgenic *GFP* gene, which points to integration of partial transformation cassettes or other unintended changes to the integration constructs. Such effects are a general feature of GE transformation methods, as reviewed by Chu and Agapito [31]. As a consequence, the generated transformants need to be assessed for the integrity of the integrated recombinant constructs, as well as the number and the location of transgenic integrations, before they may be employed in practical agricultural use [31]. Overall, in this study, an *Agrobacterium*-mediated transformation system that uses the herbicide-resistant *CP4-EPSPS* gene was developed. A low concentration of glyphosate showed lower selection efficiency but higher transformation efficiency. Transgenic sugarcane seedlings displayed no sensitivity to 0.6% glyphosate. The enhanced resistance of transgenic sugarcane seedlings to herbicide has great significance for the sugarcane industry.

## 4. Materials and Methods

### 4.1. Materials and Media Compositions

In the sugarcane genetic transformation process, all the materials and media compositions for each stage were prepared using the methods described by Wang et al. [13]. Additionally, MS (Murashige and Skoog Salts and Vitamins), sucrose, 2,4-D (2,4-Dichlorophenoxyacetic Acid), 6-BA (6-Benzylaminopurine), NAA (1-naphthylacetic acid), and Timentin were procured from PhytoTechnology Laboratories (PhytoTechnology, Lenexa, KS, USA), while phytoblend was obtained from Caisson Laboratories, Smithfield, UT, USA. Glyphosate (Roundup herbicide) used in this study was obtained from Monsanto, St. Louis, MO, USA (Table 2).

### 4.2. Plant Material and Callus Induction

Plant material of the sugarcane variety ROC22 was obtained from the Sugarcane Test and Demonstration Base of the Institute of Tropical Bioscience and Biotechnology of the Chinese Academy of Tropical Agriculture Science, Hainan, China. The immature leaf whorl found on the tops of sugarcane tillers served as the raw material for the induction of embryogenic callus, which was started within 24 h of cutting. Immature sugarcane leaf whorl transverse sections were made largely according to Bower and Brich’s [25] instructions. Sections that were transverse and 1 mm thick were taken from directly above the meristem and put on callus induction media. Callus cultures were maintained in the dark at 28 °C and sub-cultured on fresh callus induction medium every 2 weeks for a total culture duration of approximately 45 days. According to the actual callus induction status, the adjustment of 2,4-D (1–3 mg/L) and 6-BA (0–0.5 mg/L) was slightly optimized (Table 1). Light-yellow, compact calli were selected and fragmented before glyphosate killing curve testing and *Agrobacterium*-mediated genetic transformation.

### 4.3. Minimum Inhibitory Concentration of Roundup

The callus induction medium was changed with several concentrations of glyphosate (0 mg/L, 10 mg/L, 20 mg/L, 30 mg/L, 40 mg/L, and 50 mg/L) and inoculated compact calli that were made as previously mentioned for callus induction (Roundup, 41% glyphosate, Monsanto, USA). After being kept at 28 °C in the dark for 14 days, the cultures were transferred to a regeneration medium that had been altered to include the same amounts of glyphosate. They were kept at 30 °C for 20 days while receiving 14 h of light each day. For the purpose of transformation selection, the lowest inhibitory concentration of glyphosate that inhibited the growth of secondary leaves was selected.

### 4.4. Binary Vector and Agrobacterium Strain

The plant expression vector (*Bar-GFP* containing *GFP* reporter gene and *Bar* screening marker) was constructed in previous research in our lab [13]. The binary vector pCAMBIA3300 was modified to include the *bar* gene, which is driven by the CaMV 35S promoter, in between the two XhoI sites. The *npt II* gene for bacterial selection was present in the binary vector pCAMBIA3300. The Ubi1 promoter reporter gene green fluorescent protein (GFP) was inserted into pCAMBIA3300 in between the SacI and BamHI restriction sites (Figure 8). Additionally, *CP4-EPSPS* genes with XhoI restriction sites at both C and N sides were synthesized by gene biosynthesis (completed by Nanjing Detai Biological Engineering Co., Ltd., Nanjing, China). The *Bar* gene in the original vector was removed by XhoI single enzyme, and then the linearized binary vector was reconnected with the synthetic *CP4-EPSPS* gene. The recombinant plasmid with forward insertion of *CP4-EPSPS* gene was verified by sequencing. The final binary vector with CP4-EPSPS selection marker was named *CP4-EPSPS-GFP* and mobilized into *Agrobacterium* EHA105 (AC1010, Weidi Co. Ltd., Shanghai, China) via freeze-thawing with liquid nitrogen and verified by PCR analysis. The *Agrobacterium* strain was cultivated in Yeast Extract Peptone (YEP) medium, which contained the necessary antibiotics (rifampicin 10 mg/L and kanamycin 50 mg/L), and was stored in a refrigerator at −80 °C.

### 4.5. Agrobacterium Initiation

The *Agrobacterium* strain containing the *CP4-EPSPS-GFP* plant expression vector was removed from the −80 °C refrigerator and streaked on YEP solid medium (yeast extract 10 g/L, peptone 10 g/L, NaCl 5 g/L, agar 15 g/L, kanamycin 50 mg/L, and rifampicin 10 mg/L) culture at 28 °C for 2–3 days. A single colony was selected and sub-cultured overnight on fresh YEP solid medium at 28 °C. In order to initiate a starter culture of liquid initiating medium (1/5 strength MS basal salt, 30 g/L sucrose, 30 g/L glucose, and 100 uM acetosyringone), bacteria were harvested from the surface of the solid medium. The starter culture was then shaken gently (fewer than 100 revolutions) for two hours at 28 °C in the dark. The bacterial combination was then diluted until it had an optical density of between 0.3 and 0.6 at 600 nm.

### 4.6. Infection and Co-Cultivation

The compact calli that were ready for transformation were gathered and weighed (3–4 g). Each callus was arranged in a Petri dish, allowed to air dry for one hour on a sterile laboratory bench, and then moved to a 150 mL Erlenmeyer flask. After adding 100 mL of preheated (45 °C) liquid starting media (1/5 strength MS basal salt, 30 g/L sucrose, 30 g/L glucose, 100 uM acetosyringone, pH 5.4) without *Agrobacterium* to cover the calli, all of the calli were heat-shocked in an incubator for five minutes. The liquid blank initiating medium was pipetted out, and 50 mL of initiated *Agrobacterium* mixture was added in and shaken gently for 10 min at 28 °C. The calli/*Agrobacterium* mixture was sonicated for 2 min in an ultrasonic cleaner. The initiated *Agrobacterium* mixture was renewed and vacuumed for 5 min at −0.08 MPa. The calli/*Agrobacterium* mixture was shaken gently for 10 min in the dark. After the *Agrobacterium* suspension was finally pipetted out, the calli were placed on a Petri dish, the excess *Agrobacterium* suspension was wiped dry using filter paper, and the calli were allowed to air dry for about 30 min on the spotless bench. After that, the calli were placed in a fresh, empty Petri dish, covered with paraffin film, and co-cultivated with *Agrobacterium* for two to three days at 22 °C in the dark [13].

### 4.7. Resistant Plant Screening

After co-cultivation, all of the infected embryogenic calli were transferred to the recovery medium for recovery cultivation in the dark at 28 °C for one week (Table 1). Subsequently, all calli were transferred to a selection medium for 35 days in the dark at 28 °C (each of the 90 mm culture dishes placed around 15 pieces of infected embryogenic calli). After being moved to the regeneration medium, resistant calli were grown for 20 days at 30 °C under 14 h of light per day in an illuminated incubator. After being chosen, the green buds were moved to elongation medium and grown for a further 20 days at 30 °C under 14 h of light every day in the lit incubator. From each bud, a single shoot was transferred to the rooting medium and grown for 30 days at 30 °C under 14 h of light every day in a lit incubator. When the shoots reached a height of around 6–7 cm, the leaves were removed for molecular analysis, moved to pots, and then grown in a greenhouse.

### 4.8. Visualization of Green Fluorescent Protein Expression

Leaves and roots of resistant shoots were sampled for GFP fluorescence detection using a fluorescence microscope (Carl Zeiss Scope.A1, Oberkochen, Germany). Firstly, the whole infected embryogenic callus particle after co-cultivation was amplified 40 times, the number and distribution of single-cell fluorescence points on the callus were observed, and the initial infection efficiency was analyzed. After callus selection, resistant calli were amplified 10 times for the GFP (510 nm emission filter and 480 nm excitation filter) fluorescence detection followed by the estimation of growth and percentage of resistant calli.

### 4.9. Total Genomic DNA Extraction and PCR Assay

Total genomic DNA was extracted from the leaf sample of each resistant shoot. The CP4-EPSPS and GFP gene integration was found, and the lack of *Agrobacterium* contamination was verified using a PCR assay. Primers specific to the GFP, VirG, and CP4-EPSPS genes were used in the PCR experiments (Table 3). Total genomic DNA isolated from plants of the wild type was utilized as the negative control, while vector plasmid was utilized as the positive control.

The effectiveness of the sugarcane CP4-EPSPS/glyphosate transformation selection system was calculated based on the PCR assay results.

Selection efficiency (SE) = number of shoots/number of resistant shoots that are PCR positive for the CP4-EPSPS gene obtained after rooting selection.

Transformation efficiency (TE) = number of shoots/weight (gram) of the calli that are PCR positive for the CP4-EPSPS gene used for transformation.

### 4.10. Quick Stix Strips Assay

Resistant shoots were sampled and tested by Quick Stix Strips (Quick Stix Kit for CP4-EPSPS, Genesino (Dalian), Biological S&T Development Co., Ltd., Dalian, China). Approximately 10 mg of leaf tissue was taken from each resistant plant for the testing. Wild-type plant tissue samples served as a negative control.

### 4.11. Herbicide Sensitivity Assay

Tillers from the resistant shoots (asexual clones) were propagated on the rooting medium for the herbicide tolerance assay. Roundup (41% glyphosate, Monsanto, St. Louis, MO, USA) was diluted into four concentration gradients (0.2%, 0.4%, 0.6%, and 0.8%). A herbicide-free concentration (0%) was formulated as a negative control. The transgenic plants at a height of 10 cm were sprayed with different herbicide concentrations.

## 5. Conclusions and Future Prospects

In this study, an *Agrobacterium*-mediated transformation system that uses the herbicide-resistant CP4-EPSPS gene was developed. It is noteworthy that an increase in glyphosate concentration resulted in decreased regeneration capacity of the calli. Specifically, a lower glyphosate concentration had the minimum inhibitory impact on calli regeneration. A low concentration of glyphosate showed lower selection efficiency but higher transformation efficiency. Transgenic sugarcane seedlings displayed high resistance to the glyphosate. Interestingly, transgenic lines displayed a significantly decreased glyphosate herbicide sensitivity as compared to wild types. Additionally, a few transgenic plants died even at a higher concentration. Overall, this study demonstrates an effective *Agrobacterium*-mediated transformation system that utilizes the herbicide-resistant CP4-EPSPS gene as a selection marker which can be used for future purposes due to the low cost of glyphosate and efficiency of this system.

## Figures and Tables

**Figure 1 plants-13-00852-f001:**
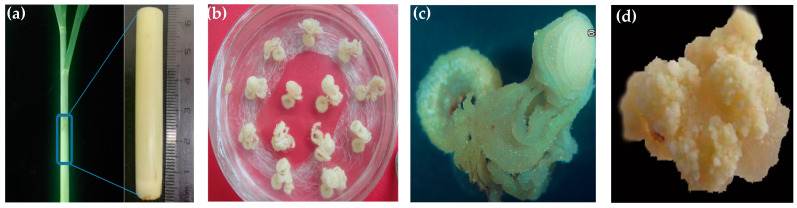
Induction of embryonic callus from immature sugarcane top stalk. (**a**) Preparation of immature top stalks of sugarcane; (**b**) transverse section transferred to callus induction medium; (**c**) mucilaginous calli formed on the transverse sections; (**d**) embryonic calli suitable for transformation after 40–45 days of induction.

**Figure 2 plants-13-00852-f002:**
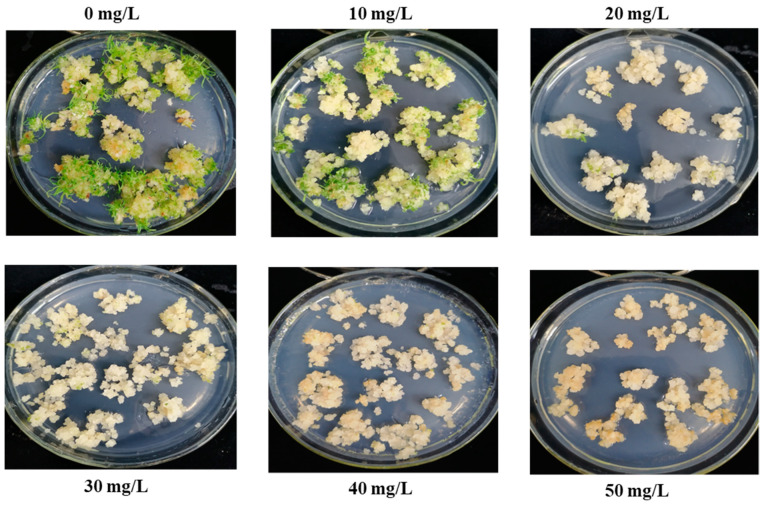
Sugarcane calli maintained on different concentrations (0 mg/L, 10 mg/L, 20 mg/L, 30 mg/L, 40 mg/L, 50 mg/L) of glyphosate.

**Figure 3 plants-13-00852-f003:**
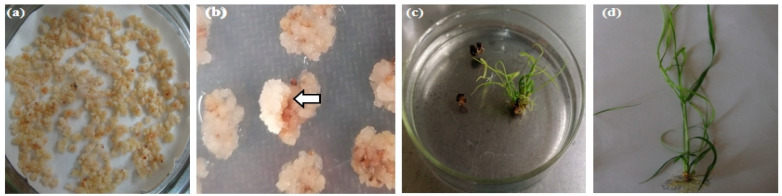
Selection of resistant shoots. (**a**) Air-dried embryonic calli; (**b**) selection of infected calli; (**c**) regeneration of resistant calli; (**d**) root emergence of resistant shoots. Note: arrow indicates resistant calli on the selection medium.

**Figure 4 plants-13-00852-f004:**
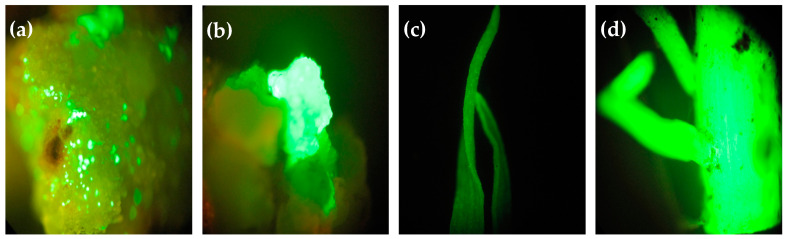
Green fluorescent protein expression in infected calli, resistant calli, and shoots. (**a**) GFP fluorescence of co-cultivated calli; (**b**) GFP fluorescence of resistant calli; (**c**) GFP fluorescence of leaf from resistant shoot; (**d**) GFP fluorescence of root from resistant shoot.

**Figure 5 plants-13-00852-f005:**
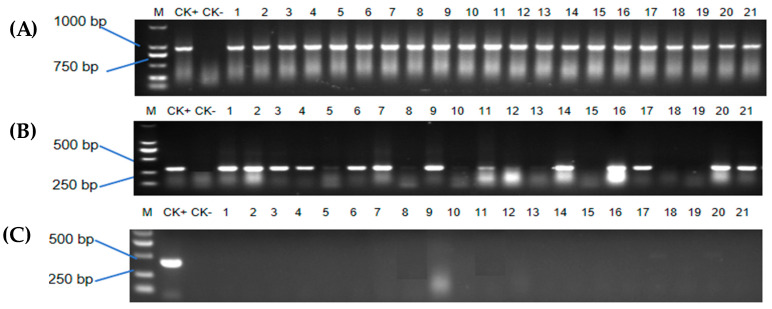
PCR verification of resistant shoots. (**A**) PCR detection of *CP4-EPSPS* gene in resistant shoots; (**B**) PCR detection of GFP gene in resistant shoots; (**C**) identification of *Agrobacterium* contamination from resistant plants via *VirG* gene amplification. M: DNA marker ladder DL2000 (HoldBio. Nanjing, China); CK+: plasmid of plant expression vector; CK−: DNA template from wild-type shoot; 1–21: resistant shoots after tissue culture.

**Figure 6 plants-13-00852-f006:**
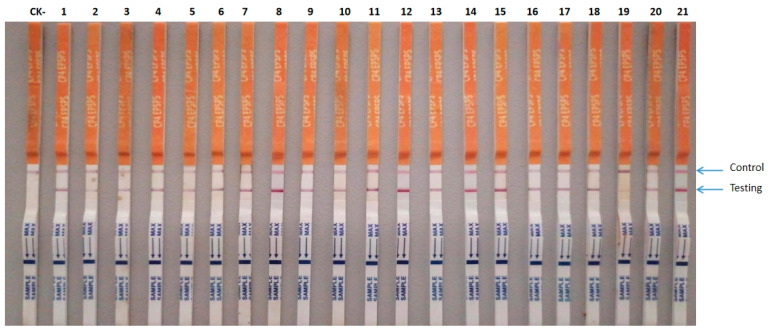
CP4-EPSPS gene expression Quick Stix Strip testing in resistant shoots. CK-: non-transformed shoot; 1–21: resistant shoots.

**Figure 7 plants-13-00852-f007:**
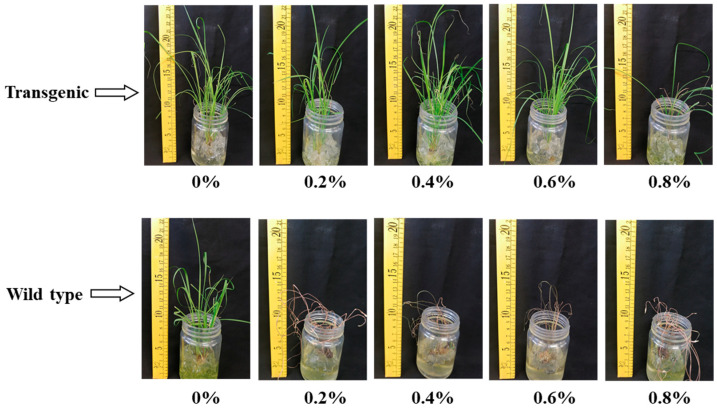
Herbicide tolerance assay of transgenic and wild-type seedlings.

**Figure 8 plants-13-00852-f008:**
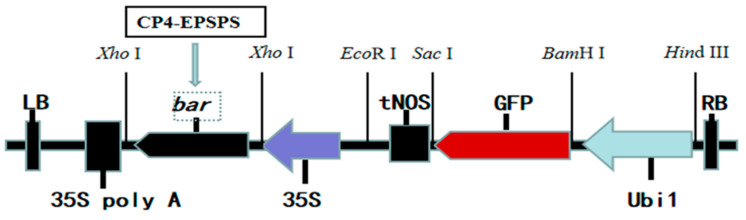
Schematic representation of the binary vector used in sugarcane transformation. Abbreviation: 35S: cauliflower mosaic virus 35S promoter; 35S poly A: cauliflower mosaic virus 35S poly A terminator; tNOS: nopaline synthase terminator; Ubi1: maize Ubi1 promoter.

**Table 1 plants-13-00852-t001:** Transformation efficiency analysis.

Glyphosate Concentration (mg/L)	Weight of Calli (g)	No. of Resistant Shoots (Line)	CP4-EPSPS Positive (Line)	SE (%)	GFP Positive (Line)	TE (Lines/g)
20	3.5	31	29	93.4	26	7.4
30	3.5	19	19	100	17	4.9
40	3.5	21	21	100	15	4.2

SE (selection efficiency) = shoots/resistant shoots that were PCR positive for the *CP4-EPSPS* gene; TE = number of shoots/weight (gram) of the calli that were PCR positive for the *CP4-EPSPS* gene used for transformation.

**Table 2 plants-13-00852-t002:** Materials and media compositions used in this study.

Medium	MS (g/L)	Sucrose (g/L)	Phytoblend (g/L)	2,4-D (mg/L)	6-BA (mg/L)	NAA (mg/L)	Glyphosate (mg/L)	Timentin (mg/L)	pH
Callus Induction	4.43	30	8	1–3	0–0.5	0	0	0	5.8
Recovery	4.43	30	8	1	0.5	0	0	300	5.8
Callus Screening	4.43	30	8	1	0.5	0	20–40	300	5.8
Regeneration Screening	4.43	30	8	0	1–2	0	10	300	5.8
Elongation Screening	4.43	30	4	0	0	0	10	300	5.8
Rooting Screening	4.43	20	4	0	0	1–2	10	300	5.8

**Table 3 plants-13-00852-t003:** List of primers used in this study.

Gene	Product Size (bp)	TM	Primer Sequence	
CP4-EPSPS	969	58 °C	Forward	GGCGACAAGAGCATCAGTCA
			Reverse	CCTCGAGACGTTCATCACGG
GFP	249	58 °C	Forward	CTGGATGAAGTGCCAGTCGG
			Reverse	TGCATGTACCACGAGTCCAA
VirG	332	58 °C	Forward	TTCGTTCCGATGCTCTATGA
			Reverse	AGGTCGTCTTTCTGCTTTCC

## Data Availability

Data are contained within the article.

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
