# Peer review of "Establishment of an Efficient Sugarcane Transformation System via Herbicide-Resistant CP4-EPSPS Gene Selection"

_plants, 2024, doi:10.3390/plants13060852_

Round 1

Reviewer 1 Report

Comments and Suggestions for Authors

Reviewer comments:

The manuscript ID (plants-2897573) entitled “Establishment of Efficient Sugarcane Transformation System via Herbicide Resistant CP4-EPSPS Gene Selection” by Wenzhi et al. I found this topic interesting, demonstrates an effective Agrobacterium-mediated transformation system that utilizes the herbicide resistant CP4-EPSPS gene as a selection marker.

But I have a few concerns related to the research article. I am asking authors to revise the manuscript carefully considering my comments for possible publication in “Plants”.

I have given my comments.

• The present investigation will be a good contribution to the genetic improvement of Sugarcane.

• Line No 176: The authors should check and correct “gglyphosate”.

• Line No 191, 230: agrobacterium must be in italics, check throughout the manuscript and correct.

• Line No 255: Authors must check and correct medium “yellow fluorescent protein (YEP) medium”.

• Authors requested to add “Lethal dose” of glyphosate for sugarcane callus.

• Authors are requested to mention what type of ladder/marker is used in gel images.

• I wish to ask authors how the current article is different from the following articles

1.               https://doi.org/10.3906/biy-1306-81,

2.               https://doi.org/10.3389/fpls.2017.01535,

3.               Racedo, Josefina, et al. "Development of a further transgenic sugarcane cultivar resistant to glyphosate herbicide." Revista industrial y agrícola de Tucumán 96.2 (2019): 59-63.

The submitted manuscript may be acceptable for publication after the revision.

Author Response

Reviewer 1:

The manuscript ID (plants-2897573) entitled “Establishment of Efficient Sugarcane Transformation System via Herbicide Resistant CP4-EPSPS Gene Selection” by Wenzhi et al. I found this topic interesting, demonstrates an effective Agrobacterium-mediated transformation system that utilizes the herbicide resistant CP4-EPSPS gene as a selection marker.

Comment: But I have a few concerns related to the research article. I am asking authors to revise the manuscript carefully considering my comments for possible publication in “Plants”.

Response: Thank you for your valuable suggestion and comments for the improvement of current manuscript. We have improved the manuscript based on comments/suggestions. All the mentioned changes have been incorporated in the manuscript. We appreciate for your warm work earnestly and hope that the correction will meet with approval.

I have given my comments.

Comment: The present investigation will be a good contribution to the genetic improvement of Sugarcane.

Response: Thank you for your appreciated comment.

Comment: Line No 176: The authors should check and correct “gglyphosate”.

Response:  We have corrected the typo error as per your suggestion.

Comment: Line No 191, 230: agrobacterium must be in italics, check throughout the manuscript and correct.

Response: We have thoroughly checked the manuscript and italicized “Agrobacterium” accordingly.

Comment: Line No 255: Authors must check and correct medium “yellow fluorescent protein (YEP) medium”.

Response: We have corrected it as per your suggestion.

Comment: Authors requested to add “Lethal dose” of glyphosate for sugarcane callus.

Response: We have added the lethal dose of glyphosate as per your suggestion.

Comment: Authors are requested to mention what type of ladder/marker is used in gel images.

Response:  We have added marker (DL2000) in the figure note as per your suggestion.

Comment: I wish to ask authors how the current article is different from the following articles

https://doi.org/10.3906/biy-1306-81,

Response:  This article (https://doi.org/10.3906/biy-1306-81) used foreign gene delivery by biolistic. It is not the best method in terms of economics and easiness of protocol to implement as well as mostly get high copy number shoots. Additionally, young emerging shoots from the transformed calli were initially screened through GUS assay. The transgenic shoots were also screened by GUS assay. After regeneration of transformed calluses, huge among of shoots emerged from each callus. Performing GUS assay for each single shoot is not economical and easy as well as time consuming. Overall, GUS assay is not advisable after transformation.

https://doi.org/10.3389/fpls.2017.01535,

Response:  The article was publish by us previously. The selection agent was pmi gene. Selection was performed via adding mannose in the tissue culture medium. The pmi gene was out of use after screening. This is not good for commercial utilization of transgenic lines. While CP4-EPSPS is a target gene for herbicide improvement.

Comment: The submitted manuscript may be acceptable for publication after the revision.

Response:  Thank you for your valuable comments. We have revised the manuscript accordingly.

Reviewer 2 Report

Comments and Suggestions for Authors

This MS is very similar to the authors' previous publication!

https://link.springer.com/article/10.1007/s12042-017-9186-7

Author Response

Reviewer 2

Comment: This MS is very similar to the authors' previous publication!

https://link.springer.com/article/10.1007/s12042-017-9186-7

Response: Thank you for your valuable comments. Bar gene is used for herbicide improvement for a lot of crop for a very long period. This system also works very well in our lab. However, this system (with other target gene’s) only used for functional identification, not for commercial utilization. Transgenic lines integrated with bar gene resist to glufosinate-ammonium. Glufosinate-ammonium is costly than glyphosate. Notably, transgenic lines integrated with CP4-EPSPS gene and other target gene is better for commercial utilization. So far no transformation reports are available which used glyphosate screening during tissue cultivation. As cooperation with breeding company, they also emphasized and asked to use the CP4-EPSPS gene for selectable marker gene. The present study is totally different from our previously published manuscript.

Reviewer 3 Report

Comments and Suggestions for Authors

Dear authors

The manuscript provides a valuable description of a new transformation method for sugarcane and an assessment of the properties of the new method which is based on an Agrobacterium method using an EPSPS-marker gene for selection of transformants. The approach is well described and can be used to facilitate and improve routine genetic engineering of sugarcane.

The manuscript would benefit from a number of minor edits as indicated in the detailed comments below.

In addition, several less editorial issues should be addressed ahead of publication:

The manuscript focuses on describing a transformation method for sugarcane that is employing Glyphosate selection, and not on the further assessment of transformed plants, which is all fine and good. However the paper indicates that 10-30% of the selected lines lack the transgenic GFP gene, which is pointing to integration of partial transformation cassettes or other unintended changes to the integration constructs. I recommend to indicate that such effects are a general feature of GE transformation methods as reviewed by Chu and Agapito (2022). As a consequence the generated transformants need to be assessed for the integrity of the integrated recombinant constructs, as well as the number and the location of transgenic integrations before they may be employed in practical agricultural use (Chu & Agapito 2022).

The manuscript also addresses the vulnerabilities of sugarcane production to the use of herbicides. A number of the effects of herbicides which are referred to in the manuscript, including the effects of Glyphosate-based herbicides and HR plants are reviewed in Schütte et al. (2017). However the discussion (L203) and conclusions (L339) just state that “The enhanced resistance of transgenic sugarcane seedlings to herbicide has great significance to the sugarcane industry in China under current climate change scenarios”. This does not explain whether the statement is meant for its direct significance (i.e. the use of Glyphosate-resistant GE-sugarcane in agriculture) or its indirect significance (i.e. the use of the Gly-based transformation system for the production of transgenic sugarcane lines with other traits which could be favourable under conditions of climate change). It also does not provide any discussion of the outcomes of the direct application of the described transformation system to generate herbicide resistant sugarcane and its use in agriculture. Thus, I suggest that the above statement in the discussion and conclusion chapters is elaborated and better explained.

Detailed comments

Abstract

L12: Consider deleting “owing to” in the sentence

L15: Consider deleting “enhancement” in the sentence

L18: Pls. revise wording; I suggest to change to
..all of the plants which are identified by PCR as transformants showed significant ...

Introduction

L28: Pls. change “in” to “to”

L31: Revise complicate to “complicated”

L32: You indicate a number of adverse impacts of climate change on sugarcane growth an productivity. Amid the list accelleration of crop development is included – explain whether this is also an adverse effect or indicate the nature of the impact.

L38: Reconsider using “detect” in the sentence and revise wording

L53: Pls. change “transgenic” to “transgenesis”

L54: change “system” to “systems”

L56: Revise wording: “high shoot regeneration and calli” – What is meant here? pls. provide a better explanation

L59: likewise; explain “high-lighting, reduced oxidation and recalcitrance”

L66: Change to: “an Agrobacterium-mediated …”
use a consistent syntax throughout the manuscript and also revise other instants where the article is missing and “agrobacterium” is in small caps (e.g. L199

67-68: Change to:
“that uses the CP4-EPSPS gene which is conferring resistance against roundup herbicide as a selection marker for sugarcane …

Results

L76: Pls. revise order: “…and excess culture medium was washed away”

L101: Insert “the” in sentence (“ containing the same glyphosate concentrations …” )

L110: Pls. revise wording (“ ..and divide apart from none resistant calli distinctly”); I suggest “Resistant calli showed bright green fluorescence in comparison with non-resistant calli”

L112: Insert “an” (revise to “which indicated an absence of chimera in transgenic shoots”)

L119: Correct mistake: change “bar” for CP4-EPSPS

L128: indicate meaning of abbreviations (CK+, CK-) in the caption of Fig 5

L137, 142: introduce abbreviation TE (transformation efficiency, according to L322)

L147: Revise wording “Testing line was appeared ..”

L156: Revise sentence:
“ At 10 days after spraying different concentrations of glyphosate, all wild-type sugarcane tillers died, while transgenic sugarcane seedlings grew healthy even when treated with up to 0.6% glyphosate.”

L169: What is “ efficient agrobacterium media” ? Mind lower caps in Agrobacterium

L171: Pls. revise to:
“By modifying the 2,4-D concentration in different stage of callus induction, ...”

L176: Pls. revise wording and spelling/delete one “d” in glyphosate (“In this research by referred the selection concentration of gglyphosate used in the genetic transformation process of maize, …”)

L188: Change “report” to “reporter”

L194: Delete “that” and “the” and use “higher” instead of high
Suggested wording: “…quick strip sticks assay positive, indicating that under higher pressure of glyphosate selection, only the shoots with CP4-EPSPS gene integration and expression can survive throughout the selection process.”

L200: change “resistant” to “resistance” (also L 337)

L202: Reconsider wording “no sensitivity”: In Chapt. 2.8. you indicate that Glyphosate concentrations higher than 0,6% have an effect on the growth of transformed sugarcane seedlings. (This also relates to conclusions L339!)

Materials and Methods

L242: delete “and saved”

L252: change “agent” to “marker”

L270: Consider changing “vortexed at 100 rpm” – the term is usually used for higher speeds of shaking

Chu, Philomena; Agapito-Tenfen, Sarah Zanon (2022): Unintended Genomic Outcomes in Current and Next Generation GM Techniques: A Systematic Review. In: Plants (Basel, Switzerland) 11 (21), S. 2997. DOI: 10.3390/plants11212997.

Schütte, G., Eckerstorfer, M., Rastelli, V., Reichenbecher, W., Restrepo-Vassalli, S., Ruohonen-Lehto, M., Wuest Saucy, A.-G., Mertens, M. (2017): Herbicide resistance and biodiversity: agronomic and environmental aspects of genetically modified herbicide-resistant plants. Environmental Sciences Europe, 29(1), 1-12. DOI: 10.1186/s12302-016-0100-y; http://enveurope.springeropen.com/articles/10.1186/s12302-016-0100-y

Comments on the Quality of English Language

included in general comments

Author Response

Reviewer 3

Comment: Dear authors, the manuscript provides a valuable description of a new transformation method for sugarcane and an assessment of the properties of the new method which is based on an Agrobacterium method using an EPSPS-marker gene for selection of transformants. The approach is well described and can be used to facilitate and improve routine genetic engineering of sugarcane. The manuscript would benefit from a number of minor edits as indicated in the detailed comments below. In addition, several less editorial issues should be addressed ahead of publication:

Response: Thank you for your worthy suggestion and comments for the improvement of current manuscript. We have improved the manuscript based on comments/suggestions. All the mentioned changes have been incorporated in the manuscript. We appreciate for your warm work earnestly and hope that the correction will meet with approval.

Comment: The manuscript focuses on describing a transformation method for sugarcane that is employing Glyphosate selection, and not on the further assessment of transformed plants, which is all fine and good. However the paper indicates that 10-30% of the selected lines lack the transgenic GFP gene, which is pointing to integration of partial transformation cassettes or other unintended changes to the integration constructs. I recommend to indicate that such effects are a general feature of GE transformation methods as reviewed by Chu and Agapito (2022). As a consequence the generated transformants need to be assessed for the integrity of the integrated recombinant constructs, as well as the number and the location of transgenic integrations before they may be employed in practical agricultural use (Chu & Agapito 2022).

Response: We have highlighted and discussed it in discussion section as per your suggestion.

Comment: The manuscript also addresses the vulnerabilities of sugarcane production to the use of herbicides. A number of the effects of herbicides which are referred to in the manuscript, including the effects of Glyphosate-based herbicides and HR plants are reviewed in Schütte et al. (2017). However the discussion (L203) and conclusions (L339) just state that “The enhanced resistance of transgenic sugarcane seedlings to herbicide has great significance to the sugarcane industry in China under current climate change scenarios”. This does not explain whether the statement is meant for its direct significance (i.e. the use of Glyphosate-resistant GE-sugarcane in agriculture) or its indirect significance (i.e. the use of the Gly-based transformation system for the production of transgenic sugarcane lines with other traits which could be favourable under conditions of climate change). It also does not provide any discussion of the outcomes of the direct application of the described transformation system to generate herbicide resistant sugarcane and its use in agriculture. Thus, I suggest that the above statement in the discussion and conclusion chapters is elaborated and better explained.

Response: We have discussed the direct and indirect effects of herbicides and HR plants in introduction section. Moreover, we have also improved the conclusion section accordingly.

Detailed comments

Abstract

L12: Consider deleting “owing to” in the sentence

Response:  We have deleted it as per your suggestion.

L15: Consider deleting “enhancement” in the sentence

Response:  We have deleted it as per your suggestion.

L18: Pls. revise wording; I suggest to change to
..all of the plants which are identified by PCR as transformants showed significant ...

Response:  We have changed it accordingly.

Introduction

L28: Pls. change “in” to “to”

Response:  We have changed it accordingly.

L31: Revise complicate to “complicated”

Response:  We have revised it accordingly.

L38: Reconsider using “detect” in the sentence and revise wording

Response:  We have revised it accordingly.

L53: Pls. change “transgenic” to “transgenesis”

Response:  We have changed it accordingly.

L54: change “system” to “systems”

Response:  We have changed it accordingly.

L56: Revise wording: “high shoot regeneration and calli” – What is meant here? pls. provide a better explanation

Response:  We have revised it accordingly.

L59: likewise; explain “high-lighting, reduced oxidation and recalcitrance”

Response:  We have explained and corrected the mistakes.

L66: Change to: “an Agrobacterium-mediated …”
use a consistent syntax throughout the manuscript and also revise other instants where the article is missing and “agrobacterium” is in small caps (e.g. L199

Response:  We have changed it accordingly.

67-68: Change to:
“that uses the CP4-EPSPS gene which is conferring resistance against roundup herbicide as a selection marker for sugarcane …

Response:  We have changed it accordingly.

Results

L76: Pls. revise order: “…and excess culture medium was washed away”

Response:  We have revised it accordingly.

L101: Insert “the” in sentence (“ containing the same glyphosate concentrations …” )

Response:  We have edited it accordingly.

L110: Pls. revise wording (“ ..and divide apart from none resistant calli distinctly”); I suggest “Resistant calli showed bright green fluorescence in comparison with non-resistant calli”

Response:  We have revised it accordingly.

L112: Insert “an” (revise to “which indicated an absence of chimera in transgenic shoots”)

Response:  We have edited it accordingly.

L119: Correct mistake: change “bar” for CP4-EPSPS

Response:  We have changed it accordingly.

L128: indicate meaning of abbreviations (CK+, CK-) in the caption of Fig 5

Response:  We have edited it accordingly.

L137, 142: introduce abbreviation TE (transformation efficiency, according to L322)

Response:  We have edited it accordingly.

L156: Revise sentence:
“ At 10 days after spraying different concentrations of glyphosate, all wild-type sugarcane tillers died, while transgenic sugarcane seedlings grew healthy even when treated with up to 0.6% glyphosate.”

Response:  We have edited it accordingly.

L169: What is “ efficient agrobacterium media” ? Mind lower caps in Agrobacterium

Response:  We have corrected it accordingly.

L171: Pls. revise to:
“By modifying the 2,4-D concentration in different stage of callus induction, ...”

Response:  We have edited it accordingly.

L176: Pls. revise wording and spelling/delete one “d” in glyphosate (“In this research by referred the selection concentration of gglyphosate used in the genetic transformation process of maize, …”)

Response:  We have edited it accordingly.

L188: Change “report” to “reporter”

Response:  We have edited it accordingly.

L194: Delete “that” and “the” and use “higher” instead of high
Suggested wording: “…quick strip sticks assay positive, indicating that under higher pressure of glyphosate selection, only the shoots with CP4-EPSPS gene integration and expression can survive throughout the selection process.”

Response:  We have edited it accordingly.

L200: change “resistant” to “resistance” (also L 337)

Response:  We have edited it accordingly.

L202: Reconsider wording “no sensitivity”: In Chapt. 2.8. you indicate that Glyphosate concentrations higher than 0,6% have an effect on the growth of transformed sugarcane seedlings. (This also relates to conclusions L339!)

Response:  We have edited it accordingly.

Materials and Methods

L242: delete “and saved”

Response:  We have edited it accordingly.

L252: change “agent” to “marker”

Response:  We have edited it accordingly.

L270: Consider changing “vortexed at 100 rpm” – the term is usually used for higher speeds of shaking

Response: We have edited it accordingly.

Chu, Philomena; Agapito-Tenfen, Sarah Zanon (2022): Unintended Genomic Outcomes in Current and Next Generation GM Techniques: A Systematic Review. In: Plants (Basel, Switzerland) 11 (21), S. 2997. DOI: 10.3390/plants11212997.

Schütte, G., Eckerstorfer, M., Rastelli, V., Reichenbecher, W., Restrepo-Vassalli, S., Ruohonen-Lehto, M., Wuest Saucy, A.-G., Mertens, M. (2017): Herbicide resistance and biodiversity: agronomic and environmental aspects of genetically modified herbicide-resistant plants. Environmental Sciences Europe, 29(1), 1-12. DOI: 10.1186/s12302-016-0100-y; http://enveurope.springeropen.com/articles/10.1186/s12302-016-0100-y

Response: Both references are cited within the text.

Round 2

Reviewer 2 Report

Comments and Suggestions for Authors

Authors have applied herbicide-resistant CP4-EPSPS gene as a selection marker in transgene transformation for sugarcane to generate gene-modified plants. This approach is better for commercial utilization which is less expensive to use. This method is of interest to researchers in the field of biotechnology and breeders. 

The potential concern lies in the similarity of MS writing to their previous publication, which also used the bar gene as a selection marker in the transformation of sugarcane.